# Markers of Subclinical Cardiovascular Disease in Patients with Adrenal Incidentaloma

**DOI:** 10.3390/medicina56020069

**Published:** 2020-02-10

**Authors:** Magdalena Szychlińska, Angelika Baranowska-Jurkun, Wojciech Matuszewski, Katarzyna Wołos-Kłosowicz, Elżbieta Bandurska-Stankiewicz

**Affiliations:** Clinic of Endocrinology, Diabetology and Internal Medicine, School of Medicine, Collegium Medicum, University of Warmia and Mazury in Olsztyn, 10-900 Olsztyn, Poland; angelika_b1990@o2.pl (A.B.-J.); wmatuszewski82@wp.pl (W.M.); kat.wolos@gmail.com (K.W.-K.); bandurska.endo@gmail.com (E.B.-S.)

**Keywords:** adrenal incidentalomas, hypercortisolism, cardiovascular risk markers

## Abstract

Due to the growing availability of imaging examinations the percentage of patients with incidentally diagnosed adrenal tumors has increased. The vast majority of these lesions are benign, non-functioning adenomas, although according to various estimates even up to 30%–50% of patients with adrenal incidentaloma may present biochemical hypercortisolemia, without typical clinical features of Cushing’s syndrome. Adrenal adenomas secreting small amounts of glucocorticoids may cause morphological and functional changes in the myocardium and blood vessels. Early stages of cardiovascular remodeling may be observed among asymptomatic patients with adrenal adenoma. Vascular changes precede the development of cardiovascular diseases and can increase morbidity and mortality in patients with adrenal incidentaloma. This risk may result not only from the traditional risk factors. Seemingly hormonally inactive adrenal tumors can indeed produce small amounts of glucocorticoids that have metabolic implications. Therefore, evaluation of patients with incidental adrenal findings presenting with subclinical cardiovascular disease seems of particular importance.

## 1. Introduction

Adrenal incidentaloma (AI) is an abnormal tissue mass in the adrenal gland with diameter ≥1 cm, found during imaging examinations performed for non-endocrine reasons. AI is a significant problem in everyday clinical practice. According to various estimates its incidence ranges from 3% to 10% and increases with age. In AI patients 50%–60% adenomas are right, 30%–40% left, and 10%–15% are bilaterally localized [1,2,3,4].

The vast majority of these lesions are benign, hormonally inactive adenomas. Nevertheless, pheochromocytoma, aldosteronism, or overt cortisol excess due to hyperfunction may be observed in patients with AI. Cortisol excess is the most frequent endocrine dysfunction among patients with AI with a prevalence ranging from 5% to 30%, primary aldosteronism has a median prevalence of 2% (range 1.1%–10%) [3,4,5,6,7,8]. Pheochromocytoma is discovered in approximately 5% of patients with AI [9]. Mansmann et al. found 41% of AI to be adenomas, 19% metastases, 10% adrenal cortical carcinoma (ACC), 9% myelolipomas, and 8% pheochromocytomas, with other benign lesions, such as adrenal cysts, ganglioneuromas, hematomas, and infectious or infiltrative lesions representing rare pathologies [10]. ACC is rare finding and can be functional or nonfunctional with regard to hormone synthesis and clinical features. Risk of ACC increases with the diameter of a tumor. ACC represent 2% of all tumors ≤4 cm in diameter and 25% of the tumors >6 cm [11,12].

The pathogenesis of AI is still unknown. Some observations showed that AI is more frequent in older patients. That led to the notion that these tumors may be a manifestation of the ageing adrenal and could represent focal hyperplasia in response to ischemic injury [13]. There are hypotheses that insulin through the mitogenic action on adrenal cortex could contribute to the development of these tumors [14,15]. Additionally, alterations in the glucocorticoid feedback sensitivity of the hypothalamic-pituitary-adrenal axis may lead to subtle but chronic trophic stimulation of the adrenals by repeatedly inappropriately higher adrenocorticotropic hormone (ACTH) levels, particularly in response to stress, favoring nodular adrenal hyperplasia [16]. Germline or somatic genetic alterations are identified only in subgroups of sporadic tumors that are mainly functioning [17,18,19].

The presence of an AI has been associated with an increased incidence of several cardiovascular factors. Patients with AI can show a high prevalence of obesity, hypertension, diabetes mellitus, glucose intolerance, and dyslipidemia [2,20]. Cortisol potentiates cardiac angiotensin II and noradrenaline responsiveness and also stimulates the local renin-angiotensin system. High cortisol levels lead to endothelial damage, activation of inflammation, increased oxidative stress, and fibroproliferation, leading to functional and structural change in the heart and blood vessels [21,22].

## 2. Objective

The aim of the study is to discuss the selected cardiovascular risk (CVR) indicators in patients with AI.

## 3. Hormonal Evaluation

Current classification of AI based on cortisol secretion includes Cushing’s syndrome (CS), autonomous cortisol secretion (ACS), possible autonomous cortisol secretion (pACS), and non-functional adrenal incidentaloma (NFAI) [6,7]. ACS, previously known as subclinical Cushing’s syndrome (SCS) or subclinical hypercortisolism (SH), is a condition of biochemical hypercortisolism in the absence of clinical symptoms of CS (Table 1) [23,24]. This definition was modified in 2016 by European Society of Endocrinology in order to distinguish CS and ACS as distinct conditions that significantly differ in morbidity and mortality rates. The risk of progression of ACS to overt CS is very low [11,25,26,27,28].

The recommended diagnostic test for hypercortisolism is 1 mg overnight dexamethasone suppression test: serum cortisol <1.8 µg/dL (50 nmol/L) excludes hypercortisolism, 1.9–5 µg/dL (51–138 nmol/L) suggests pACS, and >5 µg/dL (138 nmol/L) with the absence of CS symptoms suggests ACS [6,7]. Diagnosis of ACS is highly challenging in view of the ambiguous diagnostic criteria and the accompanying metabolic disorders that are difficult to detect. The criterion of the absence of symptoms and signs is highly controversial because it relies entirely on the clinical evaluation and personal experience of single physicians.

It is suggested that cortisol secretion is a continuum between normal serum cortisol concentration and overt hypercortisolism and shows significant variability in one patient [11,29]. Thus, the use of stiff criteria causes unavoidable diagnostic errors. In addition, the reliability of other markers used in the diagnosis of ACS, such as morning ACTH levels or 24-h urinary free cortisol is low [11,30,31,32]. Certain diseases or medications can affect the results of dexamethasone suppression tests and may hinder diagnosis of hypercortisolism [33,34].

In 1%–29% of AI patients ACS can be observed, which probably correlates with increased risk of metabolic disorders and cardiovascular disease (CVD) in this group of patients [6,7,35]. Some studies indicate that even 30%–50% of patients with AI may present with biochemical hypercortisolism with no typical clinical features of CS [36].

ACS is associated with increased CVR and several metabolic disorders and may cause structural and functional abnormalities in myocardium and arterial wall architecture [37,38,39,40,41,42,43,44,45]. These patients have higher incidence of arterial hypertension, glucose metabolism impairment ranging from insulin resistance to diabetes mellitus, lipid metabolism alterations, hyperuricemia, overweight, and obesity [23,46,47]. Latest publications demonstrate higher CVR also in patients with NFAI [48,49]. It suggests that NFAI may secrete small amounts of glucocorticosteroids, which has metabolic consequences.

## 4. Intima-Media Thickness of the Carotid Artery

Carotid intima-media thickness (CIMT) measured with ultrasonography is a marker of the advancement of generalized atherosclerosis, including coronary artery disease [50]. CIMT is a valuable indicator of a cardiovascular event risk. CIMT progression of 0.1 mm correlates with 10%–15% increase of myocardial infarction risk and 13%–18% increase of stroke risk [51,52] and is directly associated with increased arterial stiffness [53].

In 2002 the study of Taumanova L. et al. that included 28 patients with SCS showed increased CIMT value compared to the control group. In 11 (39.3%) patients with no clinical symptoms of CVD, atherosclerosis in various stages was found. Up to 87.5% of patients with SCS had multiple CVR factors, and 64% had cardiovascular impairment [54].

Recent works have again tried to determine CVR in AI patients by assessing morphological and mechanical function of endothelium.

Androulakis I. et al. examining 60 normotensive euglycemic AI patients, have demonstrated significantly higher CIMT values in cortisol secreting AI compared to NFAI and controls. NFAI patients had higher CIMT values compared to the control group. Positive predictive value of 1 mg overnight dexamethasone suppression test was demonstrated in CVR estimating (sensitivity was 79.2%, specificity was 88%) [55]. Tuna M. et al. in a group of 28 patients with NFAI and Imga et al. among 51 patients with NFAI confirmed higher IMT values compared to the control group [56,57]. Evran M. et al. have also observed higher CIMT value in 81 patients with AI compared to the control [58]. A similar conclusion was formed by Cansu G.B. et al. based on the studies conducted in 35 NFAI patients with no traditional CVD risk factors. Higher CIMT values were also showed in comparison to the control group [59]. Emral R. et al. who examined 83 patients with AI have confirmed increased CIMT in patients with NFAI. CIMT showed a positive correlation with age, homeostatic model assessment of insulin resistance (HOMA), fasting glucose, fasting insulin, total cholesterol, and fibrinogen. Fasting glucose and insulin and HOMA were significantly higher in NFAI patients than in the control group. The incidence of metabolic syndrome in the NFAI group was higher than in controls [60].

## 5. Arterial Stiffness Index

Stiffness, elasticity, and distensibility of large and medium arteries are major parameters used to assess arteries condition. Increased arterial stiffness is associated with increased cardiovascular morbidity and mortality and is an independent early risk factor for coronary artery disease and stroke in the general population [61]. Pulse wave analysis allows recording of arterial elasticity by measuring pulse wave velocity (PWV) or augmentation index (AIx) of peripheral or central arteries [62].

Cansu GB. et al. demonstrated higher PWV and peripheral and central AIx in patients with NFAI compared to the control group. PWV showed a positive correlation with total cholesterol, triglyceride, and insulin levels [59]. Similar conclusions were drawn by Akkan T. et al. who observed higher PWV and AIx adjusted for heart rate in NFAI patients. Authors have observed a negative correlation between PWV and 24-h urinary free cortisol levels [63]. In the study of Sbardella E. et al. significantly higher PWV in pACS patients was confirmed [64].

## 6. Assessment of Endothelial Function

An established method for assessing endothelial function is flow-mediated vasodilation (FMD) that measures arterial dilation in response to a transient period of ischemia. The decrease in FMD correlates with endothelial dysfunction and is associated with adverse future cardiovascular events [65,66].

In the previously mentioned study conducted by Androulakis I. et al. in 2014, lower FMD in cortisol-producing AI as compared to NFAI and controls was demonstrated. NFAI patients had lower FMD compared to the control group [55].

## 7. Echocardiography

Left ventricular hypertrophy (LVH) is an altered proportion of myocardial fibers, vessels, and intercellular substance. In addition to cardiac muscle hypertrophy, collagen type I fibers accumulation within myocardium increases muscle stiffness [67]. Left ventricular diastolic dysfunction usually precedes myocardial hypertrophy, thus being an early indicator of micro- and macrocirculatory disorders. Coronary artery disease may also play a role in the pathogenesis of LVH. Normal muscle tissue compensates for the ischemic tissue work [68]. The presence of LVH is associated with 2- to 3-fold increase of stroke, 2- to 3-fold increase of coronary heart disease, and 3-fold increase of peripheral artery disease [69]. It was demonstrated in numerous epidemiological studies that LVH is an independent risk factor for CVD morbidity and mortality [70,71].

Multiple publications assessing the amount and distribution of epicardial fat thickness (EFT) suggest its possible use as a CVD risk marker. EFT is being considered as the new risk factor for metabolic syndrome and CVD [72]. EFT positively correlates with components of metabolic syndrome and is increased in obesity, type 2 diabetes, or polycystic ovary syndrome [73]. It seems that EFT does not only reflect other risk factors, but through secretion of proinflammatory cytokines, vasoactive substances, adipokines, and growth factors may have an independent effect on cardiac muscle, coronary arteries, and formation of atherosclerotic plaque [74]. In addition, the higher the EFT volume, the smaller the coronary artery diameter and atherosclerotic plaque becomes less calcified and more unstable. This directly increases the risk of acute coronary syndrome [75,76]. A 1-SD increment in EFT is associated with a 33% greater risk of developing incident coronary heart disease [77]. Furthermore, correlation between EFT and increased left ventricular mass was established [56,78].

Echocardiography is the most important non-invasive method used for assessing cardiac morphology. Due to the presence of early morphological and functional changes in the myocardium in AI patients, it seems reasonable to perform this examination in this group.

Ermetici F. et al. in a group of 21 NFAI patients observed higher end-diastolic diameter (EDD), diastolic interventricular septum thickness IVSd, left ventricular mass index (LVMI), and lower mitral peak flow velocity in early diastole to peak flow velocity in late diastole ratio (E/A ratio) compared to the control group. Changes in echocardiography were greater in hypertensive AI patients than in the normotensive AI group. The study did not reveal a correlation between echocardiographic parameters and hormonal activity of tumors [79]. In the study of Iacobellis G. et al. in a group of 46 AI patients, greater left ventricular mass (LVM) compared to controls was observed. LVM was higher in SH than in the NFAI group. No correlation between cortisol levels and LVM was found. Among NFAI and SH patients higher EFT was observed compared to the control group [80]. In the aforementioned study of Evran M. et al. in a group of 81 subjects with AI, higher heart rate, EDD, and end-systolic diameter (ESD) were observed in patients with NFAI and SCS as compared to controls. IVSd and posterior wall thickness at diastole (PWd) in NFAI and SCS patients were increased, and isovolumetric relaxation time (IVRT) was prolonged compared to the control group. Additionally, E/A ratio was <1 in up to 79% of NFAI patients, while in the control group it was only 36% of patients. No differences in the abovementioned parameters were observed between NFAI and SCS groups [58]. In the previously mentioned study conducted by Imga N.N. et al., higher values of LVM, IVSd, Pwd, IVRT, and EFT were demonstrated in comparison with the control group. In NFAI patients with no CVD left and right ventricular diastolic dysfunction and LVH were found. Importantly, this study showed a positive correlation between EFT and LVM, as well as between EFT and CIMT [55,81].

Sbardella E. et al. performed measurements of LVMI normalized to age and body surface area (LVMI/BSA) and to age, body surface area (BSA), glycated haemoglobin (HbA1c), LDL-cholesterol, and systolic blood pressure (LVMI/b2.7) in a group of 71 patients with AI. Morphological measurements showed significantly higher LVMI/BSA and LVMI/b2.7 in pACS than in the NFA group. Incidence of LVH was also higher in pACS than in the NFA group and it included both concentric and eccentric LVH. IVSd was higher in pACS than in the NFAI group. Left ventricular diastolic dysfunction estimated with the E/A ratio, was more common in patients with pACS. Decreased E/A ratio was found in NFAI patients, but it did not reach statistical significance (*p* = 0.07). In AI patients, both mean LVMI and incidence of LVH was higher in pACS than in the NFAI group [64]. Sokmen G. et al. in a group of 30 patients with NFAI also found significantly higher LVMI, IVSd, and Pwd, as compared to the control group [82].

## 8. Final Assessment of the Available CVD Markers in Patients with AI

Due to the widespread use of imaging examinations the proportion of patients with incidentally diagnosed adrenal tumors increases. That implies the need to establish a new diagnostic and therapeutic approach in this group. By definition, non-functional adrenal adenomas that produce small amounts of glucocorticoids may have metabolic consequences and play a role in the development of morphological and functional changes in the myocardium and blood vessels. Among asymptomatic NFAI patients, early stages of cardiovascular remodeling can be observed. Complex mechanisms of epithelial damage, inflammation, oxidative stress, and fibroproliferation lead to functional and structural changes in myocardium and blood vessels. It may increase CVR and mortality in AI patients [83]. Hormonal diagnostics of hypercortisolism, in particular ACS, is difficult, thus determining CVR in AI patients and incidence of subclinical CVD seems especially important. This may help to determine further therapeutic approaches.

In Table 2 we summarize all publications included in our review. In the investigated groups of AI patients an increased CIMT was observed, illustrating morphological changes in carotid arteries expressed by intimal hyperplasia and endothelial atherosclerosis, as well as increased PWV or AIx that are the indicators of arterial stiffness [54,55,58,59,60,63,64]. Endothelial dysfunction in the form of FMD was also observed [55]. The number of publications on echocardiographic measurements among AI patients is limited. The few available studies confirmed the presence of left ventricular hypertrophy features in this subject group: increased left ventricular EDD and ESD, IVSd, PWd, or LVMI [55,58,64,79,80,81,82]. Left ventricular diastolic dysfunction usually precedes myocardial hypertrophy, thus being an early indicator of micro- and macrocirculatory disorders. AI patients had impaired left ventricular relaxation demonstrated by decreased E/A ratio and prolonged IVRT [58]. One of the studies also demonstrated decreased E/A ratio in NFAI patients, but the level of statistical significance was not reached (*p* = 0.07) [82]. Additionally, among AI patients, higher EFT volume was shown, probably being a new risk factor for metabolic syndrome and cardiovascular disease [56,80,81].

It must be highlighted that our review has a few limitations. First, summarized studies were conducted in small groups, without randomization, many without properly matched controls. Second, groups with heterogeneous inclusion criteria were compared. The vast majority of publications referred to patients with metabolic syndrome components: insulin resistance, glucose or lipid metabolism impairment, hypertension, abdominal obesity. Recent studies have also included NFAI patients without traditional CVD risk factors.

It was suggested that increased CVR in AI patients may not only result from traditional risk factors. Some studies propose that abnormal dexamethasone suppression test may be an independent CVR factor [40,42,64]. Studies conducted so far indicate that both ACS and NFAI patients have an increased cardiovascular risk [39,40,41,42,43,44,45]. This conclusion should imply an adequate therapeutic approach. There is no conclusive data on the superiority of adrenalectomy over a non-operative approach. Surgical treatment is limited to adrenal tumors with clinically significant hormone excess, and lesions suspicious of malignancy [5,6,7]. There are, however, few reports on the possible improvement of classic CVR factors after adrenalectomy in NFAI patients. In this group after surgery weight loss, reduction of blood pressure, lipid and fasting glucose levels were observed [84,85,86,87,88]. Certainly, further research is needed to verify the effect of adrenalectomy on improving CVR factors in AI.

## 9. Conclusions

Due to the widespread use of imaging examinations the proportion of patients with incidentally diagnosed adrenal tumors increases. It implies the need to establish a new diagnostic and therapeutic approach in this group.Recent studies show that AI that produce small amounts of glucocorticoids may have metabolic consequences and play a role in the development of morphological and functional changes in the myocardium and blood vessels.Functional and structural changes of cardiac muscle tissue and blood vessels may increase CVR and mortality in patients with AI.Diagnosis of AI should not only include hormonal activity but also the assessment of cardiovascular complications risk with the use of available markers.Further research is warranted to uncover benefits and drawbacks of adrenalectomy and its effect on CVR reduction in patients with AI.

## Figures and Tables

**Table 1 medicina-56-00069-t001:** Clinical presentations of Cushing’s syndrome.

1.	Low effort tolerance, weakness.
2.	Abdominal obesity with fat deposits around the face (rounded face, “moon-shaped face”), in the midsection—especially on the back of the neck (“buffalo hump”) and abdomen, supraclavicular fat pads.
3.	Skin thinning, dilatation of blood vessels on the face, plethora, wide, purple stretch marks on the abdomen, thighs, hips, breasts, armpits and elbow pits, acne, hirsutism, oily skin, easy bruising, petechiae, edema.
4.	Proximal muscle weakness.
5.	Metabolic disorders: osteoporosis, pathological fractures, glucose and lipid alterations, nephrolithiasis, fatty liver disease.
6.	Electrolyte disorders: hypokalemia, hypophosphatemia.
7.	Hypertension, heart failure, venous thrombosis.
8.	Gastric and duodenal ulcer.
9.	Menstrual disorders, decreased libido, impotence.
10.	Immunodeficiency: opportunistic infections, mycoses, severe infections.
11.	Psychiatric disorders: depression, euphoria, sleep disturbances, emotional lability, psychosis.

**Table 2 medicina-56-00069-t002:** Characteristic of included studies.

Author	Study Design	Year	Country	Study Group	Number of Patients (M/F)	Inclusion Criteria	Control Group	Age (Years)	BMI(kg/m2)	Other Comorbidities		Clinical End-Points
										DM2 (%)	Hypertension (%)	
Tauchmanova et al. [23]	Cross- Sectional ^a^	2000	Italy	SCS	28 (9/19)	Serum cortisol levels >3 μg/dL after 2 mg-DST	100 healthy controls matched for age, gender, and BMI	56	27	35.7	60.7	CIMT
Androulakis et al. [55]	Case-control ^b^	2014	Greece	Normotensive, euglycemic AI (subgroups: CSAI, NFAI)	60 (21/39)	NFAI: serum cortisol level ≤1.09 μg/dL after a LDST; CSAI: serum cortisol levels >1.09 μg/dL after a LDDST	32 healthy controls	55.6 (CSAI) 54.6 (NFAI)	27.4 (CSAI) 26.6 (NFAI)	0	0	CIMT, FMD
Tuna et al. [57]	Cross- sectional	2014	Turkey	NFAI	28	Serum cortisol levels ≤1.8 μg/dL after a 1 mg-DST; if plasma cortisol suppression was inadequate, LDST was performed.	41 healthy controls	46.7	30	-	39.3	CIMT
Imga et al. [56]	Cross- sectional	2016	Turkey	NFAI	51 (15/36)	Serum cortisol levels ≤1.8 μg/dL after a 1 mg-DST; if plasma cortisol suppression was inadequate, LDST was performed.	35 healthy controls matched for age, gender, and BMI	52	30.32	5		CIMT, echocardiogr-aphy
Evran et al. [58]	Cross- sectional	2016	Turkey	AI-subgroups pACS, NFAI; patients without DM2, HT, coronary artery disease and dyslipidemia,	81	NFAI: serum cortisol levels ≤1.8 μg/dL after 1mg DST. SCS: serum cortisol levels >1.8 ug/dL and ACTH ≤10 ug/dL after LDST	33 healthy controls matched for age, gender, body mass index	49 (SCS) 52 (NFAI)	31.8 (SCS) 28.7 (NFAI)	0	0	CIMT, echocardiogr-aphy
Cansu et al. [59]	Case-control	2017	Turkey	NFAI without traditional CVD risk factors	35 (24/11)	Serum cortisol levels ≤1.8 μg/dL after 1mg-DST	35 healthy controls	52	-	-	-	CIMT, PWV, AIx
Akkan et al. [63]	Cross- sectional	2017	Turkey	NFAI without traditional cardiovascular risk factors	35 (15/20)	Serum cortisol levels ≤1.8 μg/dL after 1mg-DST	35 healthy controls	51	29.3	-	-	AIx, PWV
Emral et al. [60]	Case-control	2019	Turkey	AI	83	-	56 controls, matched for age, gender, BMI, waist circumference, systolic and diastolic BP, smoking, concomitant disease, and medications	-	-	-	-	CIMT
Sbardella et al. [64]	Cross-sectional	2018	Italy	pACS	71 (47/24)	pACS: cortisol levels 1.9–5 μg/dL after 1 mg-DST; confirmed with 2 mg-DST; NFAI: cortisol levels ≤1.8 μg/dL after 1 mg-DST	NFAI	67	26.3	12.7	63.4	PWV, echocardiogr-aphy
Ermetici et al. [79]	Case-control	2008	Italy	NFAI without clinical or subclinical hypercortisolism	21	Cortisol levels ≤1.8 μg/dL after 1mg-DST	Controls 18 normotensive obese subjects matched for gender and body mass index (BMI) and 20 normotensive lean subjects	-	-	-	-	Echocardiogr-aphy
Iacobellis et al. [80]	Cross-sectional	2013	Italy	AI (subgroups: NFAI, ACS)	46	SCS: UFC level >70 ug/24 h; serum cortisol levels after 1 mg-DST >5 μg/dL; morning ACTH levels <10 pg/ml	30 healthy controls	62.6	28.8	-	-	Echocardiogr-aphy
Imga et al. [81]	Case-control	2017	Turkey	NFAI	70 (21/49)	Serum cortisol levels ≤1.8 μg/dL after 1mg-DST	51 healthy controls matched for age, gender, BMI	52.4	30.1	0	-	Echocardiogr-aphy
Sokmen et al. [82]	Cross-sectional	2018	Turkey	NFAI	30	Serum cortisol level ≤1.8 µg/dL after 1mg-DST	46 properly matched control subjects	51.8	34.3	3	5	Echocardiogr-aphy

ACTH—adrenocorticotropic hormone; ACS—autonomous cortisol secretion; AI—adrenal incidentaloma; AIx—augmentation index; BMI—body mass index; BP—blood pressure; CIMT—carotid intima-media thickness; CSAI—cortisol secreting adrenal incidentaloma; DM2—diabetes mellitus type 2; DST—dexamethason suppression test; FMD—flow-mediated vasodilation; HT—hypertension; LDST—long dexamethason suppression test (0.5 mg every 6 h for 2 days); NFAI—non-functional adrenal incidentaloma; pACS—possible autonomous cortisol secretion; PWV—pulse wave velocity; SCS—subclinical Cushing’s syndrome; UFC-—urinary free cortisol. ^a^ cross-sectional study—a type of observational study design. The investigator measures the outcome and the exposures in the study participants at a specific point in time. ^b^ case control study—a type of observational study design that looks back retrospectively to find the relative risk between a risk factor and an outcome.

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
