# Peer review of "Markers of Subclinical Cardiovascular Disease in Patients with Adrenal Incidentaloma"

_medicina, 2020, doi:10.3390/medicina56020069_

Round 1

Reviewer 1 Report

The authors describe a spectrum of abnormalities of adrenal adenomas-nonfunctional adrenal adenoma, possible autonomous cortisol, autonomous cortisol secretion and cushings’ syndrome and relate to previous literature for abnormalities in cardiovascular risk.

They interchange terms because the newer definitions replace the earlier literature.

I would prefer spelling out non-conventional abbreviations (AI, CS, ACS, PAC, NFAI) unless there is a space or work limitations .

Because the definition and nomenclature have changed, what used to be cushings syndrome or subclinical cushings now may be an alternate definition.  It may be that non suppressible cortisols are found in obesity or diabetes or that the elevated cortisols cause obesity or diabetes.  This review would not be able to answer that question, but it would be able to develop a hypothesis.   

 What would be helpful would be to develop a table with reference ( name, year), new definition of adrenal function as above, test criteria to define condition, number of patients (male, female), age, body max index, percent diabetes, percent hypertension, criteria for cardiovascular definitions  clinical endpoints studied (intma-media thickness, arterial stiffness index, endothelial function, echocardiography.

Once the table is generated, that abstract may provide more specific data or percentages of outcomes 

Author Response

Dear Mr/Mrs,

We found your comments extremely helpful. We took into consideration your comments and advices and developed a table with refference (author, study design, year, country, number of patients, age, BMI, comorbidities, clinical end-points). We also created a list of abbreviation to make an article more transparent.

We hope the revised manuscript will better suit the Medicina.

Sincerly,

Authors.

Reviewer 2 Report

In this review the authors aimed to summarize current knowledge about the potential of adrenal incidentalomas in increasing cardiovascular risk. Overall, the manuscript is well written, and of interest, focused on a subject that has been not sufficiently investigated.

General comments:

Introduction is very poor. In addition to the prevalence data, the authors should provide an overview of adrenal incidentalomas, the etiopathogenesis of these tumors, their classification (functioning tumors, malignant tumors, benign, non-functional tumors).

A list of abbreviation is highly recommended.

One or more tables could be useful to include the main findings from the studies cited, with other important data such as nationality of patients, age, study design, other comorbidities, confounding factors, etc.

Specific comments:

It is unclear the alternative use of the acronyms SCS, SH or ACS to indicate adrenal incidentalomas with autonomous cortisol secretion throughout the text.

Consider joining the two sentences at lines 68-70 and the two at lines 70-73.

Line 91. Please provide the full name of HOMA.

Line 110. The beginning of the sentence was written in Italic style.

Line 141. The full names of EDD and IVSd should be given here, not at lines 151 and 152, respectively.

Line 147. “Controls”, not “control”.

Line 151. “Were”, not “was”.

Lines 161-162. Please provide the full names of BSA and HbA1c.

Line 196. Consider adding “a” before “new risk factor”.

The paragraphs of “Author Contributions” and “Acknowledgments” are incomplete.

Author Response

Dear Sir/Madam,

We've founded your comments extremely helpful. We took into consideration your comments and advices. As you suggested we've created a much longer introduction with the most important data, a list of abbreviations, a table with list of cited studies and added paragraphs: Author Contributions and Acknowledgments. We've also corrected mistakes you pointed in a part "specific comments".

We hope that the revised manuscript will better suit to the Medicina.

Sincerely,

Authors.

Reviewer 3 Report

General: The authors have identified an interesting research question. “2 Markers of subclinical cardiovascular disease in patients with adrenal incidentaloma” is an interesting topic. Changes which are required are as below.

Title: looks ok

Abstract:

Looks ok

Introduction:

Add about the prevalence of unilateral and bilateral AI’s

In introduction please mention how Ais and cortisol increase CV risk. “Cortisol potentiates cardiac angiotensin II and noradrenaline responsiveness and also stimulates the local renin-angiotensin system. High cortisol levels lead to endothelial damage, activation of inflammation, increased oxidative stress and fibroproliferation, leading to functional and structural change in the heart and blood vessels.” Reference https://eje.bioscientifica.com/view/journals/eje/178/5/EJE-17-0986.xml

Please add a paragraph about  lipid profiles, waist-to-hip ratios as these are the most important  markers of cardiovascular risk in this patient population

Some studies have shown that ESR which is a marker of CV risk is increased in this patient population. This needs to be mentioned. Reference https://www.endocrine-abstracts.org/ea/0049/ea0049ep12

Tables are appropriate.

English and grammar need to be thoroughly checked throughout the manuscript.

Overall a well written narrative review.

Author Response

Dear Sir or Madam,

We've founded your review extremely helpful. We took into consideration your comments and advices. As you suggested we provided more data in the introduction. In the article we focus mainly on morphological and functional changes in myocardium and blood vessels. That is why we didn't mention the erytrocyte sedimentation rate as a marker of cardiovascular risk. Of course we understand that this marker has a huge importance in developing cardiovascular risk as well as hyperlipidemia or hypertension. In our opinion the manuscript should concentrate on the effect of adrenal adenoma as an independent risk factor of cardiovascular risk.

We hope the revised manuscript will better suit the Medicina.

Sincerely,

Authors.

Round 2

Reviewer 1 Report

The authors have performed an extensive review of the literature; however, they still need to summarize the findings. 

They spent the introduction defining CS, ACS=subclinical CS, PACS and NFAI and then combined everything into one table “Characteristics”  without separating the conditions.  When they describe in the narrative, it is unclear whether the comparison groups are NFAI or control populations or whether the patients under evaluation have CS, ACS, or PACS.  Therefore please revise the table to include the hormonal criteria using one set of definitions (that they will pick), and if testing was not performed in the study then state in the table.  Please make sure the narrative follows the table, i.e Tauchmanova does not have any criteria in the table but does in the narrative.

It would be helpful if they may summarize the difference in the studies. That is some studies show a difference in NFAI vs comparators, some only show differences in ACS and comparators.

In Methods please define cross sectional study.

In Results, section “intima-media”, the paragraph starting “Authors of publications…” should directly follow paragraph 1 after …arterial stiffness [59].

Table 1 (clinical presentation of Cushings) is unnecessary and not part of their research.

They should add a paragraph before the last paragraph stating the limitations of their study which may include: small studies, non-randomized, many without age or weight matched controls, lack of definition of hormonal status.

Author Response

Dear Sir or Madam,

We found your comments extremely helpful. We took into consideration your comments and advices. We modyfied a table and added a paragraph about limitations of our study. We would like to leave a paragraph "Recent works have again tried to determine CVR in AI patients by assessing morphological and mechanical function of endothelium." as it was, because it refers to newer works from year 2014.

We hope the revised manuscript will better suit the Medicina.

Sincerly,

Authors.

Reviewer 2 Report

The authors addressed my previous comments and significantly improved the manuscript.

Minor revisions.

In Introduction, check that you provide the full name of the acronyms on first use, not after (e.g., ACC, ACTH, HPA).

Below Table 2 please add a legend that includes abbreviations.

Author Response

Dear Sir/Madam,

We've founded your comments extremely helpful. As you suggested we provided full names of acronymes on first use.

We hope that the revised manuscript will better suit to the Medicina.

Sincerely,

Authors.

Reviewer 3 Report

The manuscript looks improved after the revisions. The authors have incorporated the reviewer suggestions and have revised the manuscript to satisfaction.

Author Response

Dear Sir/Madam,

Thank you for your response.

Sincerely,

Authors.